# Gasdermin D: A Potential New Auxiliary Pan-Biomarker for the Detection and Diagnosis of Diseases

**DOI:** 10.3390/biom13111664

**Published:** 2023-11-17

**Authors:** Ningyi Wan, Jing Shi, Jianguo Xu, Juan Huang, Delu Gan, Min Tang, Xiaohan Li, Ying Huang, Pu Li

**Affiliations:** 1Department of Clinical Laboratory, The Second Affiliated Hospital of Chongqing Medical University, Chongqing 400010, China; 2Department of Clinical Laboratory, The First Affiliated Hospital of Chongqing Medical University, Chongqing 400016, China; 3Department of Information Center, The Second Affiliated Hospital of Chongqing Medical University, Chongqing 400010, China; 4Key Laboratory of Medical Diagnostics Designated by Chinese Ministry of Education, Chongqing Medical University, Chongqing 400016, China

**Keywords:** pyroptosis, gasdermin D, biomarker, diagnostic

## Abstract

Pyroptosis is a form of programmed cell death mediated by gasdermins, particularly gasdermin D (GSDMD), which is widely expressed in tissues throughout the body. GSDMD belongs to the gasdermin family, which is expressed in a variety of cell types including epithelial cells and immune cells. It is involved in the regulation of anti-inflammatory responses, leading to its differential expression in a wide range of diseases. In this review, we provide an overview of the current understanding of the major activation mechanisms and effector pathways of GSDMD. Subsequently, we examine the importance and role of GSDMD in different diseases, highlighting its potential as a pan-biomarker. We specifically focus on the biological characteristics of GSDMD in several diseases and its promising role in diagnosis, early detection, and differential diagnosis. Furthermore, we discuss the application of GSDMD in predicting prognosis and monitoring treatment efficacy in cancer. This review proposes a new strategy to guide therapeutic decision-making and suggests potential directions for further research into GSDMD.

## 1. Introduction

Biomarkers play a critical role in medicine, particularly in disease detection and diagnosis. These measurable indicators provide valuable information about the presence, progression, and response to therapy of various diseases. Biomarkers serve as objective tools that complement traditional diagnostic methods, aiding in early detection, accurate diagnosis, and personalized treatment strategies. By analyzing biomarkers, healthcare professionals can make informed decisions regarding patient management, prognosis, and therapeutic interventions.

Broad-spectrum biomarkers offer several advantages in disease diagnosis. By targeting common molecular pathways or physiological changes shared by different diseases, these biomarkers provide a unified framework for understanding disease mechanisms and enable the development of targeted interventions applicable across various conditions. They simplify the diagnostic process by providing a single test or panel of tests that can detect multiple diseases simultaneously, saving time and resources and improving patient outcomes through early detection and timely intervention. Additionally, broad-spectrum biomarkers aid in differential diagnosis, distinguishing between similar diseases with overlapping symptoms and guiding appropriate treatment strategies.

Furthermore, pan-biomarkers have significant implications for monitoring treatment efficacy and prognostic evaluation. By measuring biomarker levels or activity before, during, and after treatment, healthcare professionals can assess the response to therapy and make informed decisions regarding treatment adjustments. This personalized approach enhances patient care by tailoring interventions to individual needs and optimizing therapeutic outcomes. Similarly, analyzing biomarker expression or functional status allows clinicians to predict disease progression, recurrence, or overall patient survival, facilitating risk stratification and guiding treatment decisions and follow-up strategies.

Several pan-biomarkers have been identified, including C-reactive protein (CRP), microRNAs, circulating tumor cells (CTCs), DNA methylation patterns, metabolites, and telomere length. These biomarkers have shown promise in various diseases, providing valuable insights into underlying pathophysiological processes.

Pyroptosis, derived from the Greek roots pyro (fire or fever) and ptosis (falling), refers to a pro-inflammatory form of programmed cell death characterized by the release of cellular contents, the activation of inflammatory responses, and eventual cell lysis [1]. This process eliminates the intracellular replication environment of the pathogen, making it vulnerable to phagocytosis by a secondary phagocyte, which then proceeds to destroy the pathogen [2]. While initially recognized as a defense mechanism against infections, pyroptosis has also been implicated in non-infectious diseases such as cancer, neurodegeneration, and cardiovascular diseases through the upregulation of inflammatory cytokines [3]. GSDMD is a key executioner protein involved in pyroptosis, encoded by the gene on chromosome 8 and belonging to the gasdermins family. It is expressed in various cell types, including epithelial cells and immune cells. The protein is emerging as a potential universal biomarker candidate due to its multiple roles in different pathological conditions. Pyroptosis plays critical roles in host defense against infections and contributes to the pathogenesis of numerous diseases, including infectious diseases, autoimmune disorders, and cancer. Here, we aim to explore the potential of gasdermin D (GSDMD) as a universal biomarker and discuss its involvement in inflammatory responses and cell death.

The activation of GSDMD leads to the formation of membrane pores, resulting in the release of pro-inflammatory cytokines and danger signals, which further amplify the immune response [4,5]. This process facilitates pathogen clearance but also contributes to tissue damage and inflammation in certain disease contexts. Given its central role in pyroptosis and involvement in diverse cellular processes, GSDMD has garnered significant attention as a potential universal biomarker.

In this review, we will delve into the molecular mechanisms underlying GSDMD activation and its functional implications in various disease settings. We will explore the current understanding of GSDMD as a potential biomarker for different diseases, including infectious diseases, inflammatory disorders, and cancer. Additionally, we will discuss the challenges and future prospects of utilizing GSDMD as a universal biomarker in clinical practice. By comprehensively examining the role of GSDMD in inflammatory responses and cell death, this review aims to shed light on the potential of GSDMD as a versatile biomarker and its implications for disease diagnosis, prognosis, and therapeutic interventions.

## 2. The Process of Pyroptosis

Pyroptosis, a highly inflammatory form of programmed cell death, is initiated by the recognition of pathogen-associated molecular patterns (PAMPs) and damage-associated molecular patterns (DAMPs) by pattern recognition receptors (PRRs). This recognition leads to the activation of caspases, which cleave gasdermins into N-terminal and C-terminal fragments, triggering pyroptosis and inflammation through plasma membrane permeabilization (Figure 1).

Pyroptosis can occur through three distinct pathways [6]. The first is the canonical inflammasome pathway, consisting of PRRs as sensor proteins, ASC as an adaptor, and caspase-1 as an effector. The second is the non-canonical inflammasome pathway, triggered by LPS and mediated by caspase-1/4/5/11. Finally, the third pathway is the caspase-3-dependent pyroptosis pathway, in which caspase-3 cleaves GSDME, resulting in the initiation of pyroptosis. The activated caspases cleave gasdermins into N-terminal and C-terminal fragments, triggering pyroptosis and inflammation through plasma membrane permeabilization (Figure 1) [6,7]. The gasdermin family comprises gasdermin A (GSDMA), gasdermin B (GSDMB), gasdermin C (GSDMC), gasdermin D (GSDMD), gasdermin E (GSDME), and pejvakin (PJVK) [8].

Among these pathways, GSDMD serves as a major effector molecule. The GSDMD gene is located at chromosome 8q24.3 and comprises 11 exons spanning 4.7 kb [9]. The protein encoded by the GSDMD gene consists of 484 amino acids and is a homolog of GSDM. GSDMs possess a flexible hinge region connecting an N-terminal cytotoxic domain and a self-inhibitory C-terminal domain (Figure 1) [10,11,12,13]. The C-terminal fragment is removed by cleavage and is believed to fold back onto GSDMD-N to inhibit its activation [14,15]. After the activation of the inflammasome, the caspase is facilitated to harbor cleavage site 272FLTD|G276 on the linker [16]. Two functional NF-κB binding sites have been reported in the GSDMD promoter and can be suppressed by melatonin in adipocytes [17]. Moreover, Nrf2 can inhibit GSDMD transcription by binding to the miR-146a promoter targeting GSDMD transcription [18]. IRF is also an essential factor in driving GSDMD promoter activation by directly engaging its binding motif [19]. The sustained inflammatory response can lead to excessive production of cell factors and immune cells, ultimately contributing to tissue damage and chronic inflammation. Shao et al. highlighted the close relationship between pyroptosis and various diseases such as chronic hepatitis, chronic nephritis, and neurodegenerative diseases, providing new insights for clinical diagnosis and treatment [20].

GSDMD plays a significant role in innate immunity and inflammation. During physiological processes, GSDMD mediates the hyperactivation of immune cells, including macrophages, dendritic cells, and neutrophils, resulting in the formation of large cell membrane pores that facilitate the release of inflammatory cytokines in response to invasive pathogens. GSDMD-mediated immune activation is tightly regulated to ensure appropriate inflammatory responses and to prevent excessive tissue damage [21,22]. GSDMD-N can bind to cardiolipin, forming transmembrane pores and ultimately leading to cell lysis (Figure 1) [13]. Compared to GSDMD-C, GSDMD-N exhibits a distinct pyroptotic morphology and is capable of inducing extensive cell death. Therefore, this review primarily focuses on GSDMD-N [23].

GSDMD-N also plays a crucial function in the physiological process of pyroptosis and is implicated in the pathogenesis of cancer and inflammation [24,25]. GSDMD is not always protective against infections; instead, it may worsen infections and pathological inflammation by potentially leading to a cytokine storm [26]. Furthermore, the role of GSDMD in cancer is complex. While it has been reported to exhibit inhibitory effects in gastric cancer, it is highly expressed and associated with increased metastasis and poor prognosis in cancers such as cervical and liver cancer [27,28]. Extensive research has demonstrated that GSDMD is widely expressed in various tissues in different diseases (Figure 2). GSDMD-N can be detected in the bodily fluids of patients affected by diseases involving pyroptosis, as it can rapidly exit into the extracellular environment though the GSDMD pore [13]. The continuous release of GSDMD closely correlates with the occurrence and development of diseases. GSDMD is the first molecule to be released into the extracellular environment and can trigger the release of IFN-β [29]. As the induction of pyroptosis diminishes, intracellular GSDMD expression and splicing cease, leading to a rapid decrease in total extracellular GSDMD levels, indicating a positive correlation between GSDMD and disease severity [30]. This characteristic makes GSDMD a potential biomarker for disease identification, early diagnosis, and the assessment of therapeutic efficacy and prognosis.

## 3. GSDMD as a Potential Biomarker for the Early Diagnosis of Infectious Diseases

Biomarkers, with their time-dependent effects, sensitivity, and early warning capabilities, have the potential to contribute to early disease diagnosis, reducing mortality rates, and improving survival and quality of life. Pyroptosis, a critical process in the occurrence and progression of major diseases [30,31,32], plays a significant role. GSDMD, as an effector molecule in pyroptosis, forms membrane pores and rapidly exits into the extracellular environment [13]. The continuous release of GSDMD closely correlates with the occurrence and development of diseases. GSDMD is the first molecule to be released into the extracellular environment and can trigger the release of IFN-β [33]. As the induction of pyroptosis diminishes, intracellular GSDMD expression and splicing cease, leading to a rapid decrease in total extracellular GSDMD levels, indicating a positive correlation between GSDMD and disease severity [30]. This highlights the potential of GSDMD for early disease diagnosis.

### 3.1. Respiratory System

Pleural effusion, the accumulation of pathological fluid in the pleural cavity, is a common clinical symptom observed in patients with respiratory diseases. Timely detection and etiological diagnosis are crucial for treatment and prognosis, as delayed identification of the cause of pleural effusion is associated with poor outcomes [32,33]. Current clinical practice exhibits suboptimal sensitivity for the early identification of pleural effusion, with limited studies demonstrating high sensitivity and specificity [34,35,36]. However, GSDMD concentrations have shown promise in differentiating various types of pleural effusion with high sensitivity and specificity, including exudative, transudative, tuberculous, malignant, and parapneumonic effusions [37]. Compared to traditional methods such as LDH and ADA detection, GSDMD exhibits higher accuracy in differential diagnosis (sensitivity: 96%; specificity: 94%). The rapid secretion of GSDMD during pyroptosis is time-dependent and induces the production of LDH, IL-1β, and IL-18 [13]. 

### 3.2. The Reproductive System

Preeclampsia (PE) is a multifactorial syndrome that occurs during human pregnancy and is characterized by high blood pressure and the presence of proteins in the urine. It can be classified into early-onset and late-onset subtypes [38]. As one of the leading causes of maternal and child mortality, prevention strategies for PE are limited. Therefore, early diagnosis plays a crucial role in predicting and preventing the disease [39]. Recent studies have shown that PE is associated with circulating factors, including inflammatory cytokines and damage-associated molecular patterns (DAMPs) [40]. Activation of the NLRP3 inflammasome has been identified as a potential mechanism underlying the initiation of PE, making GSDMD a potential marker for early detection in PE patients. The process is as follows (Figure 3): in response to danger signals such as cholesterol, urate, and glucose, interleukin-1 (IL-1) and IL-18 are upregulated, which then activate NLRP3. This activation leads to an increase in reactive oxygen species (ROS) and nitric oxide (NO), contributing to the inflammatory cascades observed in PE [41]. Under pathophysiological conditions such as hypoxia and endoplasmic reticulum stress, thioredoxin-interacting protein is triggered and found to be associated with NLPR3 inflammasome activation and pyroptosis, potentially contributing to the development of early onset PE [40]. GSDMD is positively correlated with systolic blood pressure and urine protein levels in patients with severe PE [42]. According to this study, severe PE patients exhibited upregulation of pyroptosis markers, including NLRP3 mRNA and protein, compared to healthy pregnancies (*p* < 0.0001). Additionally, GSDMD was found to be possibly associated with systolic blood pressure (R = 0.784, *p* = 0.039), diastolic blood pressure (R = 0.623, *p* = 0.017), and 24 h urinary protein excretion (R = 0.497, *p* = 0.032) [42].

### 3.3. Systemic Infectious Disease

#### 3.3.1. Sepsis

GSDMD has been extensively studied in systemic infectious diseases, including viral infection, sepsis, and disseminated intravascular coagulation (DIC).

GSDMD has been found to be involved in several viral infections, with it able to restrict the progress of infection. During pseudorabies virus infection, the NLRP3 inflammasome can facilitate GSDMD cleavage in porcine alveolar macrophage cells, resulting in pyroptosis [43]. Herpes simplex virus type 1 has also been found to activate GSDMD cleavage [44]. Rather than trigger the caspases, the Zika virus can directly cleave the peptide bond of GSDMD, leading to caspase-independent pyroptosis [45].

The cytokine storm observed in these diseases is amplified by positive feedback loops activated by NLRP3, leading to the release of IL-1, IL-18, and GSDMD. This process is regulated by the expansion of peripheral CD14+ monocytes/macrophages and an increase in IL-1β-expressing monocyte/macrophage populations in bronchoalveolar lavage fluid [46,47]. Similar positive feedback loops have been identified in sepsis, where RIPK3 and GSDMD signaling co-diffuse to amplify the inflammatory response and cytokine release in macrophages and endothelial cells [48]. The excessive production of cytokines leads to organ damage and failure, with it playing a crucial role in the mortality associated with these diseases. The expression of pyroptosis-related factors including NLRP3 and IL-1β was upregulated in HSV-1-infected mice tissue, and IL-1β was elevated in serum [44]. Research has revealed the relationship between multiple organ failure and pyroptosis in SARS-CoV-2 infection, where the binding of ACE2 to the spike protein overactivates NLRP3, leading to the activation of the renin–angiotensin–aldosterone system and complement cascade [26]. Another mechanism of organ damage is through the release of neutrophil extracellular traps (NETs), in which caspase-11 and GSDMD have been shown to play a crucial role in both sepsis-induced and SARS-CoV-2-infected models, contributing to organ dysfunction [49,50]. Based on the evidence presented above, pyroptosis is closely associated with systemic inflammation, and we speculate that its effector molecule, GSDMD, may serve as a potential biomarker in systemic diseases (Figure 4).

#### 3.3.2. Disseminated Intravascular Coagulation

Disseminated intravascular coagulation (DIC) is a severe complication of sepsis that is induced by bacterial infection and sepsis, ultimately leading to multiple organ dysfunction and death [51,52]. The pathogenesis of DIC involves the release of procoagulant tissue factor (TF) and phosphatidylserine by extracellular vesicles on monocytes, which activate platelets, neutrophils, and endothelial cells. This activation leads to the overproduction of ultra-large von Willebrand factor and plasminogen activator inhibitor 1, resulting in excess production of thrombin and fibrinolytic-suppressive DIC [51,53,54].

TF, as an initiator of the DIC process, is activated by gasdermin D (GSDMD), which is cleaved by caspase-11 and induced by intracellular lipopolysaccharide (LPS) receptors. This activation leads to pore formation, calcium influx, and phosphatidylserine exposure GSDMD activation by caspase-11, and intracellular LPS receptors trigger TF-mediated initiation of the DIC process, resulting in pore formation, calcium influx, and phosphatidylserine exposure [55,56,57]. The release of IL-1β, IL-18, and GSDMD-N caused by GSDMD pore formation exacerbates systemic inflammation and multiple organ dysfunction by hyperactivating macrophages and endothelial cells [48,58,59]. These processes play a crucial role in the development of sepsis and DIC, including the early stages and recessive early stages of DIC [48]. TF can also be triggered by histone-induced coagulopathy and thrombosis through intracellular inflammasome-independent pathways [60]. Pyroptosis-related cytokines, including outer membrane vesicles (OMVs) and IL-18, have been found to be correlated with disease severity, suggesting their potential as biomarkers [61,62,63].

In summary, the release of GSDMD, OMV, and IL-18 into the extracellular space during the process of DIC indicates the severity of the condition. The level of GSDMD in the blood circulation increases with the progression of DIC and the continuous occurrence of pyroptosis, making it a potential biomarker for the early detection, prognosis, and surveillance of DIC.

### 3.4. Liver Disease

#### 3.4.1. Non-Alcoholic Fatty Liver Disease

Non-alcoholic fatty liver disease (NAFLD) is characterized by hepatic steatosis, which is sensitized by a high-fat diet, obesity, and insulin resistance. This condition leads to a cascade of inflammation, fat necrosis, and fibrosis [64]. The pathogenesis of NAFLD is closely related to the activation of the inflammasome and subsequent inflammatory response [65]. It is believed to involve two steps.

The first step involves the up-regulation of inflammasome expression, primarily recognized by toll-like receptors. In a MyD88-dependent manner, AIM2, NLRP3 mRNA, and IL-1β protein are upregulated simultaneously [66,67]. Activated NLRP3 and AIM2 inflammasomes can trigger caspase-1 cleavage of GSDMD into N-terminal fragments, promoting cytokine secretion from hepatocytes and mediating the expression of lipogenic and lipolytic genes through the NF-ĸB signaling pathway, leading to inflammation [67,68,69]. GSDMD-N has been found to be positively correlated with non-alcoholic steatohepatitis (AUC 0.74) and shows high accuracy in differentiating patients with non-alcoholic steatohepatitis and NAFLD (AUC 0.62), suggesting its potential as a biomarker [69].

In the early stages of NAFLD, liver cell damage may not present obvious symptoms, making early diagnosis challenging. However, even mild damage to liver cells can activate pyroptosis, leading to the release of GSDMD into the extracellular space. Pyroptosis occurs earlier than other cell death modes, such as apoptosis, suggesting that the level of GSDMD in bodily fluids could serve as a sensitive biomarker for early liver cell injury.

#### 3.4.2. Viral Hepatitis

Hepatitis B virus (HBV) has been identified as a major contributor to hepatitis through the activation of NLRP3 by HBV X protein, which is triggered by elevated levels of mitochondrial reactive oxygen species (mito-ROS) [70]. This activation leads to the upregulation of IL-1 and IL-18, mediated by caspase-1 [30,71,72]. Similarly, hepatitis C virus (HCV), another common pathogen causing hepatitis, induces NLRP3 activation through glycoproteins, resulting in ASC-SPECK formation and subsequent cell death via both caspase-1 and caspase-3 pathways [73].

Accurately differentiating the natural progression stages of HBV infection is crucial for clinical treatment. Although there are ideal viral hepatitis-specific diagnostic markers, accurately distinguishing the natural progression stages of HBV infection remains a challenge currently. The circulating levels of GSDMD and IL-18 are significantly upregulated in patients with liver cirrhosis and are correlated with disease severity [30]. Notably, the activation pathway of GSDMD differs between HBV and HCV infections, with caspase-3 playing a more prominent role in inducing GSDMD activation in HCV infection compared to HBV infection [7]. 

Recent studies have highlighted the critical role of GSDMD in cell pyroptosis, a form of programmed cell death associated with inflammation. GSDMD has been shown to form pores on the cell membrane, leading to its rapid release from the intracellular compartment into the extracellular fluid. This suggests that the release of GSDMD may occur earlier than other damage-associated molecular patterns (DAMPs) and inflammatory cytokines. The formation of pores by GSDMD disrupts the cell membrane, facilitating the release of GSDMD and its associated molecules into the extracellular space. In contrast, the release of other DAMPs and inflammatory cytokines may require additional steps such as synthesis, processing, and secretion. Therefore, it is hypothesized that the release of GSDMD could serve as an early indicator of cellular pyroptosis and inflammatory response. Our previous research has provided preliminary validation of this hypothesis [37]. However, further research is needed to validate the sequential release of GSDMD compared to other DAMPs and inflammatory cytokines, as well as its dynamic changes in different diseases. Such investigations will contribute to a better understanding of the advantages and potential applications of GSDMD as a biomarker.

In conclusion, HBV and HCV infections induce NLRP3 activation and subsequent cell death pathways. The concentration of GSDMD may vary in different viral hepatitis infections, suggesting its potential as a new biomarker for distinguishing different hepatitis virus infections and monitoring therapeutic efficacy.

## 4. GSDMD Has Potential as a Novel Biomarker for the Diagnosis of Non-Infectious Diseases

### Nervous System

GSDMD has emerged as a potential novel biomarker for the diagnosis of non-infectious diseases. In the field of neurology, Alzheimer’s disease (AD) and vascular dementia (VD) are two common cognitive disorders with distinct pathological mechanisms.

AD is characterized by the formation of plaques caused by Aβ deposition and the accumulation of amyloid fibrils in the outer layer of cells [74]. This leads to the activation of scavenger receptors, which up-regulate MAPK through NF-ĸB or transient receptor potential melastatin-related 2 channels [75]. Consequently, calcium ion influx occurs, triggering NLPR3 recruitment and subsequent pyroptosis. Amentoflavone, an inhibitor of pyroptosis, has been shown to prevent Aβ 1-42-induced neurotoxicity through the AMPK/GSK3β pathway [76,77]. AD patients exhibit significantly increased levels of NLRP3, caspase-1, GSDMD, and IL-1β in peripheral blood mononuclear cells, and in vivo studies have demonstrated that systemic inflammasome-induced cells exacerbate apoptosis [78]. On the other hand, VD is a cognitive disorder secondary to cerebrovascular disease. Unlike AD, VD is believed to be initiated by AIM2-related inflammatory pathways [79,80]. However, the differential diagnosis between AD and VD remains challenging, highlighting the need for improved diagnostic methods. Recent research has suggested that the expression of GSDMD in the cerebrospinal fluid of AD and VD patients is significantly increased, albeit in different patterns. This differential expression of GSDMD holds promise for distinguishing between these two diseases and may provide better diagnostic value compared to classical biomarkers such as Tau and Aβ [81].

GSDMD shows potential as a novel biomarker for the diagnosis of non-infectious diseases. In the field of neurology, GSDMD has demonstrated differential expression patterns in AD and VD, suggesting its potential as a diagnostic tool for distinguishing between these two cognitive disorders. Further research is warranted to validate the diagnostic value of GSDMD and explore its utility in other non-infectious diseases.

## 5. GSDMD Has Emerged as a Potential Novel Biomarker for Prognosis or as a Predictive Biomarker in Cancer, including Colorectal Cancer (CRC) and Brain Lower-Grade Glioma

### 5.1. Colorectal Cancer

In CRC, GSDMD has been found to be correlated with the abundance of pro-inflammatory bacteria, such as Bacteroides, which promote the risk of cancer development. Studies investigating the genes and expression of GSDMD have shown its relationship with the prognosis of CRC [82]. Pyroptosis-related genes have been found to be closely associated with inflammation-associated genes and immune-associated genes in CRC, influencing the tumor microenvironment. This provides a basis for early diagnosis and medication use in CRC based on a prognostic model [83]. The downregulation of GSDMD in CRC tissues has been significantly related to tumor stage, nodal stage, lymph node invasion, and clinical stage, indicating that low expression of GSDMD is an independent unfavorable factor for the prognosis of CRC [84,85,86]. Additionally, the subcellular localization patterns of GSDMD have been found to be related to the development of CRC, with positive cytoplasmic expression indicating a lower probability of distant metastasis and positive nuclear expression indicating deeper infiltration depth. High expression of GSDMD has been identified as an independent favorable factor for prognosis in CRC [84].

In brain lower-grade glioma, pyroptosis-induced inflammation has been found to trigger robust antitumor immunity [75]. IL-1, a downstream molecule of pyroptosis and proinflammatory cytokines, creates a microenvironment conducive to tumorigenesis and is a risk factor for prognosis [76,77]. Pyroptosis-related gene expression, including NLRC4 and NLRP12, has been found to be increased in glioma [78]. Activation of NLRC4 and NLRP3 inflammasomes in microglia and astrocytes, as well as the upregulation of IRF3 and IRF7, contribute to the poor prognosis of glioma [79]. The activation of NLRP3 can trigger the downregulation of miRNA-214, which inhibits glioma cell proliferation and migration [77]. Both NLRC4 and NLRP3 have been found to be upregulated and associated with poor prognosis in glioma. GSDMD, as an effector of pyroptosis, has the potential to be a biomarker for prognosis in glioma. Prognostic models based on pyroptosis genes have revealed the biological function and clinical correlation of pyroptosis, with high expression of GSDMD indicating poor prognosis [80]. However, further in vivo or clinical verification studies are needed to fully understand the process of how pyroptosis and GSDMD affect cancer. GSDMD shows promise as a novel biomarker for prognosis in cancer, including CRC and brain lower-grade glioma. Further research is needed to validate its diagnostic and prognostic value and to explore its utility in other types of cancer.

### 5.2. Brain Lower-Grade Glioma

Pyroptosis-induced inflammation has been identified as a representative anti-tumor immune function [87]. IL-1, a downstream molecule of pyroptosis and proinflammatory cytokines, creates a microenvironment conducive to tumorigenesis, increasing the migratory and proliferative capacity of glioblastoma cells and serving as a risk factor for prognosis [88,89]. The expression of pyroptosis-related genes, including NLRC4 and NLRP12, has been found to be increased in glioma [90]. Subsequently, activation of NLRC4 and NLRP3 inflammasomes in microglia and astrocytes, along with the upregulation of IRF3 and IRF7, contributes to the transcription of caspase-11 and ultimately leads to NLRP3 activation through the JAK/STAT pathway [91]. Activation of NLRP3 triggers the downregulation of miRNA-214, which inhibits glioma cell proliferation and migration by targeting caspase-1 [89]. Both NLRC4 and NLRP3 have been found to be upregulated and associated with poor prognosis in glioma, with high expression of NLRC4 suggesting lower overall survival compared to low NLRC4 expression [91], indicating that NLRC4 may serve as a potential biomarker for glioma prognosis and suggesting a correlation between pyroptosis and glioma development. Consequently, GSDMD, as an effector released into the blood, may serve as a potential biomarker for prognosis. Lin Shen et al. developed a prognostic model based on pyroptosis genes, revealing the biological function and clinical correlation of pyroptosis, with AUCs at 1 year, 2 years, and 3 years of 0.670, 0.734, and 0.723, respectively [92]. As an effector of pyroptosis, GSDMD is involved in the progression of glioma. Analysis has shown that GSDMD is highly expressed compared to normal tissue and is associated with poor prognosis, with AUCs at 1 year, 2 years, and 3 years of 0.781, 0.754, and 0.680, respectively [93]. The process by which pyroptosis and GSDMD affect cancer is still not fully understood, and further in vivo or clinical verification studies are urgently needed.

### 5.3. Pancreatic Cancer

Pancreatic cancer is a highly lethal malignancy with an extremely poor prognosis [94]. Most patients with pancreatic cancer remain asymptomatic until the disease reaches an advanced stage, and there is currently no standardized screening protocol for high-risk individuals [95]. Recent studies have highlighted the role of pyroptosis in tumorigenesis, with widespread amplification of pyroptosis-related genes (PRGs) such as AIM2, NLRP3, GSDMD, and IL-18 observed in pancreatic cancer [96]. The subsequent activation and secretion of proinflammatory cytokines, including IL-1 and IL-18, contribute to oncogenic mutations, tumor promotion, and angiogenesis through inflammatory responses [97], suggesting that pyroptosis may serve as a prognostic factor for cancer outcomes (Figure 5). Among the gasdermin protein family, GSDMD is the only member expressed in pancreatic tissue [98]. Intracellular levels of GSDMD are significantly increased during the early stages of pancreatic cancer development, but further investigation is required to determine if GSDMD levels are elevated in bodily fluids or the systemic circulation.

A prognostic model based on four cancer stem cell-related gene signatures has been developed for pancreatic ductal adenocarcinoma, revealing that high expression of the GSDMD gene is associated with unfavorable outcomes (AUCs: 1 year: 0.722; 2 years: 0.862) [99]. Due to inherent resistance to apoptosis in most tumors, including dysregulation of death receptors, negative regulation of post-receptor signaling, and imbalanced apoptotic-related proteins, conventional pro-apoptotic therapies have limited efficacy in pancreatic cancer, which is known to exhibit resistance to agents such as 5-FU, gemcitabine, and taxanes [100]. As an alternative form of programmed cell death, pyroptosis has emerged as a promising target for cancer management. GSDMD, cleaved by caspase-4/11, has been implicated in the antitumor function of CD8+ T cells through upregulation, while GSDMD deficiency has been shown to reduce the cytolytic capacity of CD8+ T cells [101,102]. Analysis of pyroptosis levels in the tumor immune microenvironment and their prognostic implications has suggested that low pyroptosis levels are associated with improved enrichment scores of CD8+ T cells and five-year survival rates, with AUCs of 0.73, 0.69, and 0.77 at 1-, 2-, and 3-years, respectively, providing a potential strategy and predictive biomarker for immunotherapy [96]. While most current research has focused on the gene level, demonstrating the key role of GSDMD in tumor progression and its potential as a therapeutic target, further in vivo experiments are warranted to validate the influence of GSDMD concentration in the blood and its predictive value for prognosis.

### 5.4. Other Cancers

GSDMD, a critical component of the pyroptosis pathway, has been found to have potential diagnostic value in certain other tumors (Table 1). In non-small-cell lung cancer (NSCLC), co-expression analysis has revealed a relationship between GSDMD and EGFR/Akt signaling. Downregulation of GSDMD in NSCLC has been shown to inhibit tumor proliferation through an intrinsic mitochondrial apoptotic pathway [103]. Conversely, the downregulation of GSDMD in gastric cancer (GC) has been found to significantly promote tumor proliferation by accelerating S/G2 cell transition. This effect is mediated by the activation of extracellular signal-regulated kinase, signal transducer and activator of transcription 3, and phosphatidylinositol 3 kinase/protein kinase B signaling pathways. These findings suggest that GSDMD may serve as a protective prognostic factor for GC [104].

A pan-cancer analysis investigating the impact of GSDMD on prognosis in kidney renal clear cell carcinoma, rectum adenocarcinoma, skin cutaneous melanoma, adrenocortical carcinoma, and liver hepatocellular carcinoma has revealed that GSDMD may play different roles at different stages of tumor development. This analysis highlights the potential of GSDMD as a prognostic marker in various cancers [93].

The role of GSDMD in cancer is complex and context-dependent, leading to diverse disease outcomes across different tumor types. In this section, we aim to provide potential explanations for the observed heterogeneity in GSDMD-associated disease outcomes.Genetic alterations within tumor cells can significantly impact the expression and functionality of GSDMD. For instance, mutations in genes involved in the regulation of GSDMD, such as NLRP3 or caspase-1, may lead to dysregulated pyroptosis and altered disease outcomes [106]. Additionally, somatic mutations in GSDMD itself have been reported in certain cancers, suggesting a direct link between GSDMD mutations and disease progression [107].

It is important to consider that GSDMD-mediated pyroptosis is not the sole cell death mechanism involved in cancer. Crosstalk between pyroptosis, apoptosis, necroptosis, and autophagy pathways can occur, leading to complex interactions and potentially influencing disease outcomes [108,109]. The interplay between GSDMD and these alternative cell death pathways may vary across different cancers, contributing to the observed heterogeneity.

Cancer is a highly heterogeneous disease, both at the genetic and phenotypic levels. Within a single tumor type, there can be substantial intratumoral heterogeneity, resulting in diverse subpopulations of cancer cells with distinct molecular characteristics [110]. It is plausible that certain subclones within a tumor possess specific alterations or dependencies on GSDMD-mediated pyroptosis, which could contribute to the observed variability in disease outcomes. The tumor microenvironment plays a crucial role in the behavior of cancer cells. Variations in the composition of immune cells, stromal cells, and cytokine profiles within the tumor microenvironment can influence the expression and activation of GSDMD. Consequently, the extent of GSDMD-mediated pyroptosis may vary, leading to different disease outcomes. Furthermore, the presence of specific immune cell subsets, such as tumor-infiltrating lymphocytes or myeloid-derived suppressor cells, may modulate the immune response and contribute to disease heterogeneity [111].

Different tumor microenvironments can influence the expression and activation of GSDMD through various potential mechanisms; immune cells, including T cells, macrophages, and dendritic cells, are present in the tumor microenvironment. These immune cells secrete cytokines such as interferons and interleukins, which can regulate the expression and activation of GSDMD. The presence of specific immune cell types or cytokines may enhance or inhibit GSDMD-mediated cell pyroptosis [112]. Persistent inflammation is often observed in certain tumor microenvironments and is associated with tumor development and progression. Inflammation can activate inflammatory signaling pathways, such as the NLRP3 inflammasome, leading to increased expression and activation of GSDMD. This can result in higher levels of cell pyroptosis and impact disease outcomes. Tumor cells themselves can secrete factors known as tumor-associated factors (TAFs) that modulate the tumor microenvironment. These factors can directly or indirectly influence the expression and activation of GSDMD. For example, certain TAFs may affect the transcriptional level or post-translational modifications of GSDMD, thereby impacting its function [113]. The tumor microenvironment is characterized by high levels of oxidative stress, which is associated with tumor development and treatment resistance. Oxidative stress can alter the intracellular redox balance and influence the activation and cell pyroptosis of GSDMD. This may contribute to the differential expression and functionality of GSDMD in different tumors [114].

While our study focused on elucidating the role of GSDMD in cell pyroptosis and its potential as an early biomarker for inflammatory diseases, we acknowledge that the specific mechanisms underlying the diverse disease outcomes in different cancers were not extensively discussed. Factors such as the tumor microenvironment, genetic alterations, and interplay with other signaling pathways likely contribute to the varying disease outcomes observed in different cancers. The expression and activation of GSDMD could be influenced by these factors, leading to differences in disease progression and prognosis. Future studies should aim to explore the specific molecular mechanisms through which GSDMD contributes to different cancer outcomes. This could involve investigating the interaction between GSDMD and other molecules or pathways known to be involved in cancer development and progression. Additionally, a comprehensive analysis of GSDMD expression patterns and their correlation with clinical parameters in different cancer types would provide valuable insights into its prognostic significance.

## 6. Evaluation of Treatment

There is a growing body of evidence suggesting that GSDMD plays a dual role in disease progression, acting as both a disease initiator and suppressor. It promotes a more inflammatory response by releasing cellular content such as DAMPs, chemokines, and cytokines. However, GSDMD can also have anti-tumorigenic effects. Modulating pyroptosis, either by inhibiting or inducing it, may offer a novel therapeutic approach for regulating disease development. Therefore, monitoring the concentration of GSDMD in bodily fluids could provide a means of evaluating treatment efficacy.

Several potential drug targets related to pyroptosis have been found to control the activation of GSDMD and combat inflammation. These include caspase inhibitor VX-765 [115], NLRP3 inhibitor NEK7 [116], JAK/STAT inhibitor ruxolitinib [117], and disulfiram targeted for NPL4 [118]. These findings offer promising directions for treatment. On the other hand, there are also pyroptosis inducers that promote pyroptosis for cancer treatment, such as DPP8/9 inhibitor [119], nanoparticles [120], and chemotherapy drugs taxol, Tanshinone IIA, and metformin PPVI [121]. Positive feedback loops involving pyroptosis and immune response have been found to trigger robust antitumor immune responses [122].

Based on the aforementioned research, the expression of GSDMD is closely associated with disease severity, making it a potential biomarker for monitoring treatment efficacy. A 30-gene inflammasome-based risk score (IRS) system, incorporating IL1B-, CASP-1-, IL18-, GSDMD-, and inflammasome-regulated genes, has been proposed to predict the response to immune checkpoint inhibitors (ICIs). High-risk IRS is associated with increased T cell-inflamed activity, indicating a positive prediction for ICI response [123]. Another prognostic model, based on the pyroptosis-related gene score (PRG-Score), has been developed to guide immunotherapy, as it is a crucial marker for identifying individuals who are responsive to ICIs. The high PRG-Score group exhibits a higher proportion of malignant cells expressing GSDMB+ and GSDMD+. High PRG-Score is positively correlated with pyroptosis and is an independent valuable signature for invasive bladder cancer [124].

GSDMD holds promise as a predictive biomarker for therapeutic response in cancer treatment (Table 2). However, most current studies are based on model development, and their reliability needs to be validated in patient populations. Furthermore, in vitro and in vivo experiments are necessary to further elucidate the underlying mechanisms and refine the models.

## 7. GSDMD Inhibitors: Therapeutic Potential, Benefits, and Adverse Effects

GSDMD inhibitors have gained significant attention in recent years as a potential therapeutic agent for treating inflammation-related diseases. The uncontrolled activation of pyroptosis has been implicated in the pathogenesis of various diseases, making GSDMD an attractive target for drug development.

### 7.1. Disulfiram

Disulfiram, a drug commonly used to treat alcohol addiction, has recently been identified as a specific inhibitor that restricts the formation of pores on membranes by GSDMD [144]. This drug indirectly targets GSDMD by inhibiting NET formation through the inactivation of the NLRP3/Caspase-1/GSDMD pathway [50,145]. While disulfiram has shown efficacy in treating several inflammatory diseases including systemic lupus erythematosus [146], septic shock [144], and atherosclerosis [147], a study on GSDMD-independent inflammation has demonstrated that NLRP3 inflammasome activation cannot be suppressed by disulfiram when the activation overwhelms the protection offered by GSDMD inhibition [148]. Additionally, its half-life of 7.3 h in plasma is translated into other metabolites that have not been extensively studied, limiting its application. Although disulfiram has considerable potential and wide application prospects, research on its use in treating inflammation is still in its early stages. There is limited research on its clinical efficacy, side effects, and adverse outcomes, and further in-depth research is required.

### 7.2. Necrosulfonamide

Necrosulfonamide is a recently reported necroptosis inhibitor that targets the kinase domain-like protein, which has been shown to downregulate the expression of NLRP3 and GSDMD [149]. This compound directly affects GSDMD via binding Cys191 on GSDMD to prevent the oligomerization and pore formation or indirectly inhibits GSDMD by suppressing the mRNA expression and protein expression of pyroptosis-related genes [150,151]. Necrosulfonamide plays a protective role in many inflammatory diseases, including inflammatory bowel disease [149], acute liver failure [152], sepsis [151], and allergic rhinitis [153]. However, in a study on osteoblasts, the overexpression of caspase-1, GSDMD, and NLRP3 was found to abolish the effects of necrosulfonamide [150]. Although various experiments in vivo or in vitro indicate the broad application prospect of necrosulfonamide, the side effects of necrosulfonamide remain unclear, and the benefits of its clinical application are still unknown.

## 8. Conclusions

In conclusion, GSDMD holds great potential as a diagnostic, monitoring, and therapeutic target for various diseases. Its concentration in bodily fluids shows promise as a detection biomarker for early disease detection. GSDMD is widely expressed in tissues and can be used to diagnose and predict disease progression. Additionally, GSDMD may serve as a prognostic marker and efficacy monitoring indicator for chemotherapy and immunotherapy. Further research is needed to fully understand the regulatory mechanisms and signaling pathways of GSDMD in different diseases. This will help identify potential therapeutic targets and develop more effective treatment strategies. Additionally, studies should focus on validating the clinical utility of GSDMD as a diagnostic and prognostic biomarker in larger patient cohorts and diverse disease populations. In terms of detection methods, efforts should be made to optimize and standardize mass spectrometry and next-generation sequencing techniques for quantifying GSDMD expression. The quantitative analysis of GSDMD in bodily fluids could provide valuable insights into disease-specific aberrations and enable real-time monitoring of disease conditions. Moreover, GSDMD has the potential to be used as an efficacy assessment tool for therapeutic interventions.

Future studies should explore the feasibility of using GSDMD as an efficacy assessment tool for therapeutic interventions. By monitoring changes in GSDMD expression before and after treatment, clinicians may be able to evaluate the effectiveness of different therapies and tailor treatment plans accordingly. Furthermore, it will be important to investigate the potential of GSDMD modulators as therapeutic agents. Developing specific inhibitors or activators of GSDMD could provide new avenues for targeted therapy, potentially improving patient outcomes and reducing side effects.

## Figures and Tables

**Figure 1 biomolecules-13-01664-f001:**
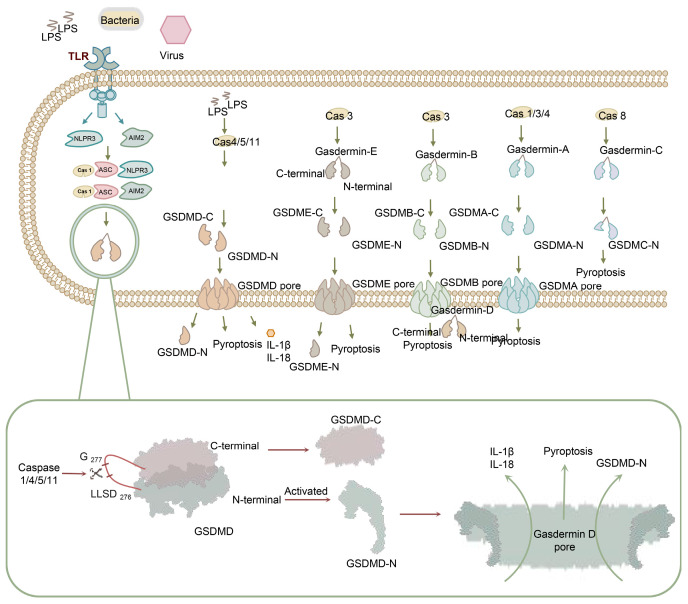
The process of pyroptosis involving gasdermin A (GSDMA), gasdermin B (GSDMB), gasdermin C (GSDMC), gasdermin D (GSDMD), and gasdermin E (GSDME). The toll-like receptor recognizes LPS, bacteria, and viruses and leads to the activation of caspases. Caspase cleaves gasdermins into N-terminal and C-terminal fragments. GSDMD-N binds to cardiolipin, forming transmembrane pores and causing cell lysis and cytokine release. Other GSDMs can also lead to pyroptosis.

**Figure 2 biomolecules-13-01664-f002:**
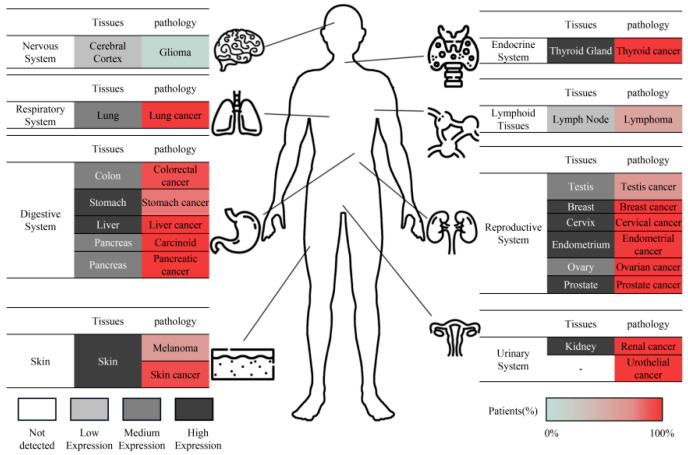
Overview of GSDMD expression according to the Human Protein Atlas [31]. The grayscale representation in the figure indicates the weighted annotation of cellular protein levels based on GSDM transcript levels in normal tissues, extracted from the HPA RNA-seq dataset within the Human Protein Atlas. Meanwhile, the color-coded depiction represents the weighted annotation of the proportion of patients exhibiting moderate to strong cytoplasmic and membranous immune reactivity in cancer tissue. Immunohistochemistry staining using 3,3′-diaminobenzidine substrate linked to horseradish peroxidase is employed to visualize the tissue sections, which are then counterstained with hematoxylin to enhance the microscopic features.

**Figure 3 biomolecules-13-01664-f003:**
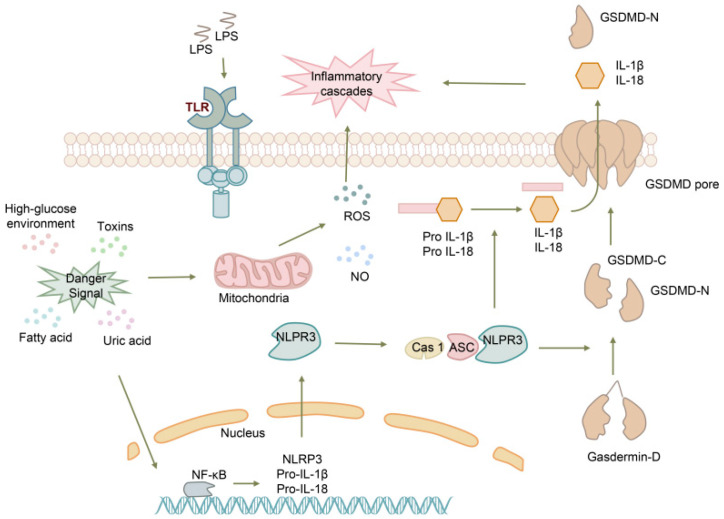
The process of GSDMD in the pathogenesis of preeclampsia. High sugar environments, toxic substances, uric acid, and fatty acids act as risk factors, transmitting information to the cell nucleus, activating the NFκB signaling pathway to increase the expression of NLRP3, Pro-IL1β, and Pro-IL-18, while also activating mitochondria to release ROS and NO. TLR can recognize LPS, transmit signals to NLRP3, and activate GSDMD and cytokines. Activated cytokines, GSDMD, ROS, and NO together activate the inflammatory cascade reaction.

**Figure 4 biomolecules-13-01664-f004:**
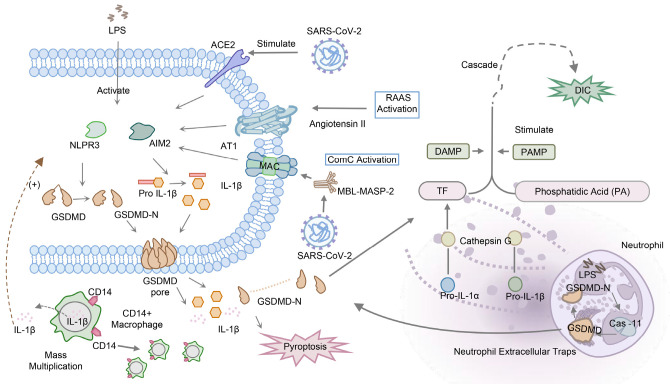
The process of GSDMD in the pathogenesis of systemic onfection. SARS-CoV-2 can overactivate NLRP3 through the binding of the spike protein and ACE2 while also causing the activation of the adrenaline–angiotensin–aldosterone system and complement cascade. SARS-CoV-2 can also activate MAC through the MBL-MASP-2 pathway, leading to the binding of NLRP3 and ultimately activating GSDMD. Activated GSDMD and caspase-11 have also been found in neutrophil extracellular traps, activating the DIC cascade reaction through the activation of tissue factor and phosphatidic acid.

**Figure 5 biomolecules-13-01664-f005:**
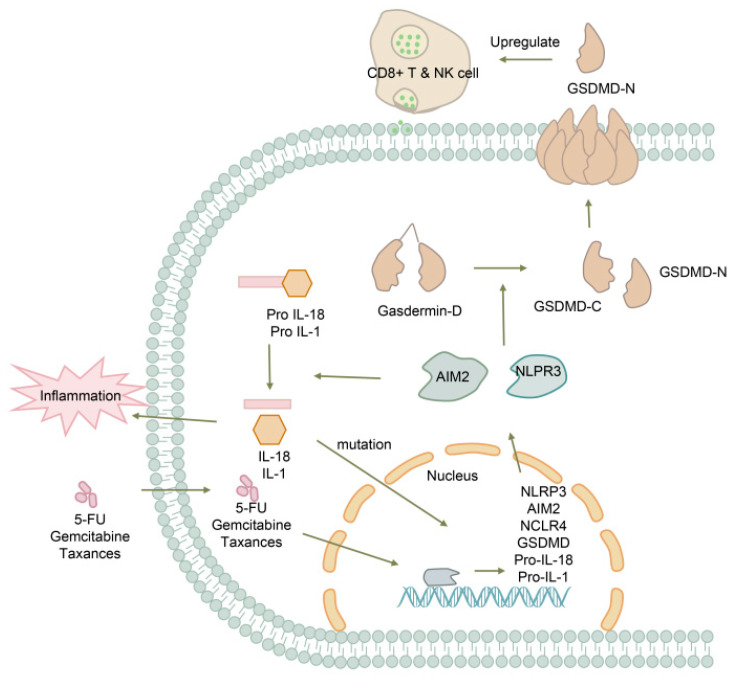
The role of GSDMD in pancreatic cancer pathogenesis. Chemotherapy drugs such as 5-FU can enter the cell nucleus and regulate the expression of NLRP3, AIM2, NCLR4, GSDMD, and cytokine precursors, thereby increasing the activation of GSDMD and upregulating the function of CD8+ T cells and NK cells.

**Table 1 biomolecules-13-01664-t001:** GSDMD and disease–patient studies for prognosis.

	Diseases	Expression	Sample Source	Sample Size	Technique	Prognosis	Results	Refs
Nervous system	Brain lower-grade glioma	↑	Human tumor tissues	530	Data analysis	Poor	The GSDMD model had good predictive ability in the survival of LGG (AUC for 1 year: 0.781; 3 years: 0.754; 5 years: 0.680)	[93]
Endocrine system	Adrenocortical carcinoma	↑	Human tumor tissues	92	Data analysis	Poor	The GSDMD model had good predictive ability in the survival of ACC (AUC for 1 year: 0.740; 3 years: 0.708; 5 years: 0.714)	[93]
Respiratory system	Lung adeno- carcinoma	↑	Human NSCLC tissues	162	Immuno- histochemistry	Poor	The GSDMD protein level was an independent, poor prognostic factor of LUAD	[103]
	Liver hepato- cellular carcinoma	↑	Tumor tissues	-	Data analysis	Poor	GSDMD was highly expressed in LIHC, leading to a poor prognosis (AUC for 1 year: 0.569; 3 years: 0.520; 5 years: 0.604)	[93]
Digestive system	Rectum adeno- carcinoma	↓	Tumor tissues	-	Data analysis	Good	High expression of GSDMD improved the survival time of patients with READ (AUC for 1 year: 0.513; 3 years: 0.408; 5 years: 0.269)	[93]
	Colorectal cancer	-	Human peripheral blood and tumor tissues	178	Immuno- histochemistry	Poor	High expression of cytoplasmic GSDMD was an independent favorable indicator for prognosis	[84,86]
	Gastric cancer	↓	Tumor tissues	61	Western blotting and qRT-PCR	Poor	GSDMD expression was decreased in GC (specificity: 0.735; sensitivity: 0.672), promoting the proliferation of tumors	[104]
Urinary system	Kidney renal clear cell carcinoma	↑	Human tumor tissues	-	Data analysis	Poor	(AUC for 1 year: 0.513; 3 years: 0.408; 5 years: 0.269)	[93]
Reproductive system	Breast cancer	↑	Human tumor tissues	108	Immuno- histochemistry	Good	GSDMD was negatively related to the histopathologic grade, the tumor size, the clinical stage, the possibility of lymph node metastasis, and the risk of death	[105]

Studies for GSDMD in prognosis of cancers. ↑ means GSDMD was found to be up-regulated in tissues, while ↓ means down-regulated in tissues.

**Table 2 biomolecules-13-01664-t002:** GSDMD and disease–patient studies for detection and differential diagnosis.

	Diseases	Expression	Sample Source	Sample Size	Technique	Results	Refs
Nervous system	Alzheimer’s disease	3.19 ± 0.55 ng/mL	Human cerebrospinal fluid	60	ELISA	GSDMD was differentially expressed in AD and VD	[8]
	Vascular dementia	1.35 ± 0.34 ng/mL	Human fluid	60	ELISA	GSDMD was differentially expressed in AD and VD	[8]
	Neonatal hypoxic–ischemic encephalopathy	↑	Human peripheral blood	9	qRT-PCR and Western blotting	The NLRP-3/caspase-1/GSDMD axis is required for microglia pyroptosis and activation	[120,125]
	Anti-NMDAR encephalitis	7.21 ± 3.53 ng/mL	Mice peripheral blood	-	qRT-PCR and Western blotting	GSDMD-mediated pyroptosis exacerbates the inflammatory response and liver damage and is regulated by CD38	[126]
Respiratory system	Chronic obstructive pulmonary disease	↑	In vitro	-	Western blotting and qPCR	Nicotine exposure increased the expression levels of GSDMD in an epithelial cell line, which may be associated with the progression of COPD	[127]
	Severe acute respiratory syndrome coronavirus 2	↑	Human blood monocytes	60	Immunoblot and ELISA	GSDMD was significantly elevated in COVID-19 patient plasma	[128]
	Acute respiratory distress syndrome	↑	Human blood monocytes	40	ELISA	The expression of GSDMD-N protein in PBMC was significantly increased and negatively correlated with PaO_2_/FiO_2_ in ARDS patients	[129]
Digestive system	Intestinal Behçet’s syndrome	16.6% ± 1.9%, 9.8% ± 1.3%, ↑	Human intestinal tissues	30	Immunohistochemistry and qRT-PCR	GSDMD was significantly increased in the intestinal tissues of patients with IBS	[130]
	Inflammatory bowel disease	↑	Human intestinal mucosal tissue	15	Immunofluorescence	The function of GSDMD depends on its subcellular location: cytoplasmic GSDMD improves prognosis while GSDMD of the nucleus promotes tumor invasion and metastasis	[131]
	Liver fibrosis	↓	Rat hepatic stellate cells	-	qPCR and Western blotting	GSDMD was decreased in hepatic stellate cells	[27]
Urinary system	Diabetic kidney disease	↑	Human kidney tissues	43	Immunohistochemistry and Western blotting	GSDMD is closely related to tubular injury	[132,133,134]
	Lupus nephritis	↑	Tumor tissues	43	Immunohistochemistry and Western blotting	GSDMD was strongly induced and cleaved	[135]
Reproductive system	Ovarian cancer	↑	Human ovarian tissue	578	Data analysis	GSDMD is differently expressed among several histotypes of epithelial ovarian cancer	[136]
	Preeclampsia	↑	Serum and placenta tissue	10	Immunofluorescence	GSDMD and its signaling pathway proteins are significant pathophysiologies	[40,42]
	Recurrent spontaneous abortion	↑	Decidual tissue	105	Immunohistochemistry, Western blotting, immunofluorescence, and ELISA	GSDMD was upregulated in the decidual tissues of URSA patients	[137]
	Myocardial infarction	↑	Rats’ myocardial tissues and mice’s cardiomyocytes	41	ELISA	GSDMD was significantly elevated in chronic MI patients	[138,139,140]
Muscle tissue	Myocardial reperfusion injury	↑	Mice’s heart tissue	-	Immunofluorescence and immunohistochemistry	I/R caused gasdermin D expression increase in vivo and in vitro	[141]
	Atherosclerotic	↑	In vitro	100	Immunofluorescence, Western blotting, and RT-PCR	Ox-LDL was found to promote the expression of GSDMD-N	[142]
	Diabetic cardiomyopathy	↑	Rats’ cardiac tissue	60	Western blotting, histology staining, and fluorescent staining	AIM2-siRNA alleviated GSDMD-N-related pyroptosis in H9c2 cardiomyoblasts	[143]

Studies for GSDMD in prognosis of cancers. ↑ means GSDMD was found to be up-regulated in tissues, while ↓ means down-regulated in tissues.

## Data Availability

The datasets used or analyzed during the current study are available from the corresponding author upon reasonable request. All data and materials as well as software applications support the published claims and comply with field standards.

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
