# Peer review of "Gasdermin D: A Potential New Auxiliary Pan-Biomarker for the Detection and Diagnosis of Diseases"

_biomolecules, 2023, doi:10.3390/biom13111664_

Round 1

Reviewer 1 Report

Comments and Suggestions for Authors

The manuscript "GSDMD: A Potential Novel Panbiomarker for Disease Detection and Diagnosis" presents interesting information, but I believe it needs to be improved considerably before publication.

- In the abstract it is necessary to define what GSDMD means

- In the introduction you use "a highly inflammatory form of programmed cell death" three times, which is very repetitive.

- I consider that in the introduction it is necessary to define better and more extensively what Pyroptosis is. Subsequently enter GSDMD

- The structure of section two seems inadequate and difficult to follow. I consider that it would be clearer for the reader if it were in the following order:

1. Pyroptosis and the role of gasdermins in this process

2. Structure of the GSDMD gene (which does not appear), protein structure and regulation (which does not appear)

3. Biological function of GSDMD and role in pyroptosis

4. Role of GSDMD in diseases

- The reference in figure 2 is not appropriate according to the journal

- Something that I consider extremely important for the article. You mention that GSDMD is elevated in several pathologies, however, in clinical practice I do not understand how it would be useful. If it rises in so many pathologies, what information does it give me to establish a diagnosis? I think it would be very ambiguous. Could it be applied for a differential diagnosis? Could it help the physician distinguish if it is an infectious process, cardiovascular, nervous system disease or cancer? Can it guide the physician to direct some type of treatment? I say this because otherwise I don't see how it could be considered a diagnostic biomarker. It would be more of a prognostic and follow-up marker once the diagnosis is established.

- Section 3.3, instead of being titled Systemic Infectious Disease, should be called COVID-19 since it is the only disease talked about in that section.

- A section should be added talking about GSDMD inhibitors (e.g. disulfiram) and their application, benefits and side effects in some diseases. Are there activators? Would they be easy to design?

Comments on the Quality of English Language

In general, fine

Reviewer 2 Report

Comments and Suggestions for Authors

The manuscript “GSDMD: A Potential Novel Pan-biomarker for Disease Detection

and Diagnosis” written by Ningyi Wan et al. presents an interesting review about involving of Gasdermin D in disease detection and diagnosis.

In general, the data are detailed, but the presentation itself needs some revision. Some details should be added.

Here, some comments.

1)                       GSDMD – acronym should not be in the title  -  Gasdermin D: A Potential Novel Pan-biomarker for Disease Detection and Diagnosis

2)           Figure 1. GSDMD, GSDMA, GSDMB, GSDMC, GSDME – should be spelled - Gasdermin D (GSDMD), Gasdermin A (GSDMA) … If using acronym in figures, at first time, it should be spelled.

3)             “Pyroptosis, a form of programmed cell death mediated by gasdermins, particularly GSDMD, is widely expressed in tissues throughout the body”. – correct please

4)            “Caspase-1/4/5/11 cleaves GSDMD into N-terminal and C-terminal fragments”.  – add more information about GSDMD protein as a Figure - the site of cleavage, the size of fragments, stricture…

5)          “Our previous research has provided preliminary validation of this hypothesis (Li et al., 2021)”. – add the reference number

Comments on the Quality of English Language

Quality of English is OK

Reviewer 3 Report

Comments and Suggestions for Authors

The main concern for this review is the proposal of gasdermin D, GSDMD as a biomarker for cancer detection and prognosis. This is based on a handful of studies in different cancers, and lack a rigorous analysis of the sensitivity and specificity of GSDMD for detection of each cancer type, as well as treatment prognosis for each cancer type.

Comments on the Quality of English Language

The quality of English language is good.

Round 2

Reviewer 1 Report

Comments and Suggestions for Authors

I consider that the manuscript entitled "Gasdermin D: A Potential Novel Pan-biomarker for Disease Detection and Diagnosis" has improved considerably. However, I am not convinced by the answers to my comment 6 (If it is elevated in so many pathologies, what information does it give me to establish a diagnosis? Could it be applied for a differential diagnosis? Could it help the physician to distinguish whether it is an infectious process, a cardiovascular disease, nervous system disease or cancer? Can it guide the physician to direct some kind of treatment?). Because of this, I consider, since you mention that Gasdermin D is more of an auxiliary diagnostic tool, that the title should be changed to "Gasdermin D: A Potential New Auxiliary Pan-biomarker for the Detection and Diagnosis of Diseases" or something similar.

Comments on the Quality of English Language

Fine

Author Response

Dear reviewer:

Thank you for your thorough review and valuable suggestions.

We have incorporated your feedback and revised the title to "Gasdermin D: A Potential New Auxiliary Pan-biomarker for the Detection and Diagnosis of Diseases" We believe this adjustment more accurately reflects the potential of Gasdermin D as an auxiliary diagnostic tool, aligning with your suggestion for the title modification.

Thank you for your time and consideration.

Sincerely,
Pu Li
